# Camel Genetic Resources Conservation through Tourism: A Key Sociocultural Approach of Camelback Leisure Riding

**DOI:** 10.3390/ani10091703

**Published:** 2020-09-20

**Authors:** Carlos Iglesias Pastrana, Francisco Javier Navas González, Elena Ciani, Sergio Nogales Baena, Juan Vicente Delgado Bermejo

**Affiliations:** 1Department of Genetics, Faculty of Veterinary Sciences, University of Córdoba, 14014 Córdoba, Spain; ciglesiaspastrana@gmail.com (C.I.P.); sergionogalesbaena@gmail.com (S.N.B.); juanviagr218@gmail.com (J.V.D.B.); 2Department of Biosciences, Biotechnologies and Biopharmaceutics, Faculty of Veterinary Sciences, University of Bari ‘Aldo Moro’, 70121 Bari, Italy; elena.ciani@uniba.it

**Keywords:** animal-based tourism, camelback riding, quality service, customer satisfaction, return intention probability, biodiversity conservation

## Abstract

**Simple Summary:**

Consumers’ environmental awareness have appeared on the scene and are bringing nature-based tourism demands to the rise. Related practices have led to camels being used as an income generator for tour companies and locals in exotic appealing destinations. The long-term viability of the respective breeds markedly depends on the fact that the principal income sources for camel breeders in some regions are tourist activities. The present research was intended to outline what are the social and cultural factors that can act as tourism attractions and satisfaction conditioning elements within this particular tourism enterprise. Staff performance, sociospatial motivating factors, diverse and humane close interaction, camel behavior and performance, sociotemporal context, and positive previous experience, are the principal dimensions affecting experiential consumption in camelback riding tours. The understanding of such experiential consumption enables tour operators to customize its camel saddlery offers based on the diverse patterns influencing the buying behavior and general satisfaction of the customers. In its broadest sense, responsible and sustainable camel-based tourism will create positive impacts for destination economies and provide long-term conservation opportunities both at a natural and cultural level.

**Abstract:**

Camels are exotic elements, which can be comprised within adventure travel companies promoting ecotourism activities. Such recreations contribute to sustainable livelihoods for local communities and educational empowerment towards nature and its conservation. At present, some local camel breeds’ survival reduces to this animal-based leisure industry and its reliability to perform and promote customized services accurately. By conducting an on-site questionnaire to customers participating in camelback riding tours, we assessed the motivational factors affecting participation, satisfaction, and loyalty in this tourism segment that may have made it socially differentiated. The sixfold combination of staff performance, culture geography, diverse and humane close interaction, camel behavior and performance, sociotemporal context, and positive previous experience involves the elemental dimensions that explain customer satisfaction and return intention probability within this entertainment business. Customer knowledge is essential for stakeholders to build personalized riding experiences and align profits with environmental sustainability and biodiversity mainstream concerns into their everyday operations. In turn, domestic camel tourist rides could be managed as a viable path to nature conservation by helping endangered local breeds to avoid their functional devaluation and potential extinction.

## 1. Introduction

Animal-based tourism is an emerging business around the world, which lets holidaymakers closely view or interact with animals. The interaction between animals and tourists may be the basis on which to support conservation and/or an educational purpose [1,2]. The animal species involved, the type of encounter, its surrounding, and animal welfare subjective perception play a pivotal role in the experience making potential or authenticity for visitors [3]. Besides, certain species and naturalness leisure activities are commonly found to be iconic and/or unique for some destinations and often linked to the livelihoods of local communities [4].

Tourists engage with a country or region’s culture not only for its past heritage but for its practices in a living context [5]. The reconsideration of attractions as huge promotors for local communities’ economic development and sustainable enhancement [6], also requires that inhabitants undergo substantial cultural and social changes, both positive and negative, to adapt to the new demands derived from these touristic resources. The impacts occurring via tourism may lead to changes in societal structure and values in different traditional daily practices [7]. Additionally, tourists’ daily decisions may impact animals and local residents in travel destinations. This, in turn, may imply foreign and local people attitudes towards animal welfare may evolve as a result of these animals or species used in the creation of tourism experiences being simply conceived as economic resources [8].

Increasing social awareness on animal welfare issues has made animal-human touristic encounters face harsh constant criticism and global concerns over the years about the wider impacts on animals’ freedom violation [9]. For this reason, tourism companies involved in this leisure experiences have started to implement innovative techniques (behavioral/environmental enrichment, proper feeding and nursing). In other words, animal welfare has progressively become a critical criterion for tour operators when selecting their suppliers and aiming at satisfying tourists’ demands, as the latter are more conscious about positive animal well-being support when choosing to book an animal-based tourism service [10].

For camel-based tourism, research briefings on the social side and economic impact of this subsector worldwide are noticeably scarce. Donlon et al. [11] and Seifu et al. [12] underlined the relevance of camel wrestling in Turkey and camel ecotourism in Botswana, respectively, and its potential as a heritage resource within this animal-based tourism enterprise. The investigation of the sociocultural attractions for foreign visitors of these animal-associated spectacles in Turkey was carried out by Çalışkan [13]. Instead, no comprehensive quantitative data regarding public perspective about camel-human leisure interactions, the most relevant factors that lead customers to make use of these activities and those potentially conditioning their general satisfaction and return intention probability, are available.

To the knowledge of the authors, the present paper constitutes the first holistic approach in this applied-research field, largely focusing on domestic camel rides in Spain. Originally from the nearest African coast, the one-humped camel (*Camelus dromedarius*) arrived in the Canary Islands around 1405 accompanying the first human expeditions of the European colonization of the islands [14,15]. Their rapid adaptation to the climate and orographic conditions of this emplacement led to the expansion of the camel herd throughout the archipelago, although its presence was and still is higher in the southern areas of Gran Canaria, Tenerife, and the whole insular territory of Fuerteventura and Lanzarote [16,17]; less numerous and isolated herds could be found along the Iberian Peninsula, mainly in southern areas. After several decades of isolation and health-based importing restrictions from other geographically closed camel populations, dromedaries in Canary Islands could differentiate themselves as a distinct breed through evolution and genetic drift [18].

For centuries, Canarian camels participated in multiple agricultural labors, military operations, and as beast of burden at short and long distances [16]. According to historical chronicles [19,20], the opening of island ports to Atlantic traffic in the late 1800s led to the evolvement of an active tourism network that completely changed the functional destination of these animals in the islands a few decades later. The mechanization of agricultural works and transportation means from the last third of 20th century made its census suffer a dramatic reduction in rural areas [21] and being this species progressively adapted to the transport of tourists instead [15]. Fortunately, tourism rise and expansion in the 1990s resulted in a population recovery. Since then, these animals have a cardinal role in the tourism industry, which is the principal income source for local camel breeders [18] apart from the European Union live animal market [22,23]. Other emerging but still minoritarian productive niches of this breed, which lack phenotypic characterization and selection programs, are milk, meat, and wool [24,25].

The current census of this camel breed is estimated on 1200 individuals, and it is included in the Spanish Official Catalogue of Livestock as an endangered autochthonous breed since 2012 by the Order AAA/251/2012 of the Spanish Ministry of Agriculture, Food and Environment. As it constitutes the only Spanish and European traditional camel population, this distinctive breed deserves functional revaluation for selective breeding with conservation purposes [21,26,27]. With this objective, an in-depth assessment of camel tourism dimensions is fully required given its far-iconic tourism attractiveness.

The present research aims at identifying which are the most important quality service attributes or demand features that influence customer general satisfaction and return intention probability in camel-riding tourist walks. Comprehensive data gathering animal attributes (behavioral features, welfare status, and global performance), staff attitudes, tourist motivations, and overall satisfaction have been recorded. The applicative results are intended to drive such popular recreational ridings in the most sustainable way as possible, both for the welfare of the animals themselves and their long-term viability as an ecotourism alternative, fortifying avenues in camel production and biodiversity conservation strategies [28]. Additionally, the exploration of consumer attitudes and preferences towards camel-riding experiences can provide insights to further reorientation and research on other leisure events such as educational activities and assisted therapy in which the camel is an element of the interactive experience.

### 1.1. Theoretical Background

#### 1.1.1. Riding Camel Conformation/Performance and Customer Satisfaction

Camels are capable to travel long distances in the desert and other sandy areas on minimum energy expenditure even under extreme environmental conditions [29]. However, when used in recreational rides, these animals may have to walk on tarred, hard-surfaced grounds that can injure their soles and make the animal lame. It is, therefore, very important to provide additional cares to their feet and legs, since rideability or suitability as riding subjects in animals is notedly influenced by the rhythm and quality of the gaits, apart from personality traits [30].

In this sense, saddle camels are sought to be well-balanced, tough, energetic, and stable on their limbs so they are light and agile in its movements, and thus, the rider can find camel riding easy and feel comfortable, both at slow and fast gaits. Their bones are finer in comparison to those camels used as beasts of burden and their feet are medium-sized, large enough to support both their own weight and the rider’s and small enough to move with agility on sandy soils. Although these conformational features may be desirable for the tasks that they are required to develop, animals lacking some of them can be also trained to become riding camels, with the inconvenience that they may offer a less comfortable interaction with the rider, especially over long journeys [31].

Concerning comfort of sitting on camel’s back, as their gravity center is supposed to be about 15 cm above and behind the elbow, it is preferable for the rider’s position to be in front of the hump for better control over the animal. Furthermore, due to the particular style of walking of camels as an adaptation for sandy environments [26], instead of trying to resist the camel swaying, it is preferable to keep the balance while rocking along with the camel movements, not against them. For novel riders, a position behind the hump is preferred since it could provide the rider with a higher sense of security [31]. An alternative option may be using the more complex English saddle, iron-made and composed of a pair of lateral sitting platforms both with seatbelt.

In addition to the aforementioned physical features, riding camels may be indirectly selected considering the dimensions of their personality or for the psychological attributes being involved in the learning of the tasks that they are required to be engaged in, during training sessions and tourist walks [27]. Effective training methods both inherited from tradition and backed by science in working animals, aligned with the camel’s natural learning abilities, will allow for the most efficient way of physical, physiological, and psychological stresses withstanding. Hence, visitors would be provided with a worthwhile experience as a result of an outstanding suitable workout.

#### 1.1.2. Tourism Geography and Culture

The territorial nature of tourism has become an object of study by social sciences for a wider understanding of the key geographical factors that operate in the territory when creating tourist development processes. It is indisputable that tourism is a spatial phenomenon as the set of activities or experiences it comprises cannot take place in the absence of physical space [32], where a set of objects, both natural and cultural, acts as a tourist attraction. Territorial development patterns are based on the physical and regional geography and competitiveness and reflect special attributes of these resources around which tourism is built [33].

Although for architectural tourism, stakeholders focus development policies around the site or sites in question, rural or nature-based tourism may present a dispersing effect [34]. At a geographical range, this last type of tourism comprises numerous relatively small-scaled sites where different rural lifestyle activities are offered [35]. However, some natural resources (i.e., plant or animal species or peculiar orography) are exclusive (autochthonous) of concrete places for pure ecological reasons. Such exclusive place-specific attractions might have the potential to increasingly improve destination attractiveness and bring economic support for nature conservation if well managed.

On the other hand, the symbolic and representational dimension of tourism or the notion of space as a product of a certain type of social practices provides a better comprehension of the processes of the social and cultural transformation of a community in its encounter with the tourist community [36]. In this perception of tourism as a force for social and cultural change, the tourist dynamic is subjected to the permanent search for new forms of organization of space and territorial configuration to preserve the cultural richness of destinations, while answering sustainability concerns and fulfilling the interests of the communities that inhabit these territories and traveler motivations [34]. In the same context, tourism geography helps to integrate the processes of the social construction of spaces, social relations, mental perceptions and representations, experiences, and interests of both receiving communities and tourists in the exercise of ordered territorial planning of the tourist phenomena [37].

#### 1.1.3. Staff Attitudes and Quality Service

Several research studies have inquired into the direct connection between employees’ positive and negative behaviors and customers’ overall satisfaction with service encounters [38]. They all conclude that the higher the required standards staff attitudes and productivity reach, the more appropriate care and effort costumers will be served with. In this scene, companies must promote their employees’ understanding of their role in business success through investing time, money, and effort on their development, engagement, and positivity. Such adequate training and proper resources’ disposal will, in turn, be translated into the fulfilment of customers’ needs and goals and also be reflected in staff manners towards the company, coworkers, and customers [39]. At last, consumers’ assertive judgment may influence the financial growth and stability of the service company as their positive satisfaction leads to their trust and willingness in promoting the enterprise and its products to others as well. Moreover, consumer bolstering drives further opportunities when seeking for investors and suppliers, which will only embrace partnerships with business organizations that fairly succeed in customers’ expectations attainment [40].

#### 1.1.4. Customer Knowledge in Domestic Animal Tourism

In the last decades, biological local resources (wild and domestic flora and fauna) have become major appealing subjects within ecotourism. The sustainability of ecotourism involves responsible traveling to natural areas, environmental sustainability, and social responsibility towards nature preservation [41]. These natural attractions act as direct financial support for nature self-sustaining and conservation and for some territories they represent a high percentage of the gross domestic product [42].

Specific consumers attempt to escape from massive destinations and get involved in activities that let them disconnect from their daily routine and enrich their environmental consciousness [43]. Some of these distractions promote close interactions with domestic animals in fenced pens or vast territories under controlled conditions [1,44], being tourists provided with exceptional and memorable experiences. They order for interactive encounters that allow them to satisfy a combination of social, recreational, and personal emotional needs at the same time they contribute to financial empowerment for local people. In the purchasing election, they prioritize those destinations and travel suppliers that are known to follow animal welfare’s management standardized practices and nature sustainable exploitation’s principles according to personal previous experience(s), previous experience(s) of others, and media promotion [45,46]. The emotional capabilities of animals per se contribute to create an hedonic encounter, which improves experience memorability and then can add psycho-pedagogical values to the activity [44].

Demographic structure compels all age groups and a uniform distribution of both genders. Regarding customer social profile within this tourism brand, they are generally medium and superior educated with high purchasing powers. This latter finding could explain their willingness to pay more for tourism companies with recognized financial promotion of environmental awareness, nature conservation, and locals’ economic empowerment [47].

#### 1.1.5. Tourism and Animal Ethics

Tourists are increasingly offered nature experiences involving animals as the prime focus both ex situ or in the animal’s natural habitat. In this context, camels are sometimes forced to perform in ways that are uncommon and improper for their species that rises environmental and social concerns. This situation has driven this entertainment industry to reassess its specific guidelines for an integrative sustainable approach not only aiming enjoyment but also emphasizing educational intentions and corresponding to animal rights [46]. New inquiries into the moral acceptability of animal use for human pleasure and the proper cares they need to be supplied with if used for recreation activities should be therefore addressed. In this regard, Hall and Brown [48] state that the viability of tourism operations strongly depends on the extent in which animal welfare concerns are present within tourism industry.

Nonetheless, despite large literature can be consulted on the welfare of animals immersed in the major animal sector (laboratories and factory farming), respective work for animals used for entertainment purposes continues to be a minority. The small number of respective studies in this field are brief theoretical summaries and species-contextualized researches that use the concept of animal welfare in a general manner [49,50,51,52]. Furthermore, since the maintenance of high levels of welfare markedly depends on animal species and also on idiosyncratic temperament features, complex ethological characterizations and animal welfare official regulations are widely available for many species but remains scarce for others (i.e., camels) [53]. In this scenario, until more objective data are available for practical use, it becomes crucial to take into account that not only forced works by trainers but also the disability of these and the riders to interpret animal behavior and/or needs and sometimes anthropomorphism attitudes (the misattribution of human traits, emotions, or intentions to nonhuman entities) are detrimental to optimal animal welfare [54].

Conclusively, the real impact of these activities thus is an emerging contemporary research topic within this specific consumption experience and tourism policies that needs further efforts [55]. In case of absence of a species-specific ethogram, environmental enrichment programs and well-being subjective appraisal of animals before, during and after participating in leisure activities can be regularly performed by multistep evaluation general protocols [56]. Ethical operations will be reached through management decisions, the impacts this industry has on animals minimized and the social reputation of these interactive experiences improved toward an ethics-focused oncoming in tourism planning [57].

## 2. Materials and Methods

### 2.1. Study Sample

Our animal sample comprises 8 dromedaries (6 males and 2 females; aged between 4 and 32 years), and the English type saddles were used for tourist rides. All the consumers that booked and enjoyed a camel ride with independence from their country of origin (Table 1) during the research period (1 July to 30 September 2019) were asked to voluntarily fulfil an on-site questionnaire.

### 2.2. Study Area

The present empirical inquiry is based on the investigation of a particular case, which may be suitable for extrapolation to similar enterprises as they all share homogeneous framework elements and strategies, i.e., camel breeding in nationally protected arid and semiarid areas and their exploitation in touristic activities [58]. Protected areas are a key part of programs to conserve biodiversity and ecosystems as they are conceived as large terrain extensions where multidisciplinary management strategies can be executed both for protecting natural resources (physiographic regions, biotic communities, genetic heritage, and unimpaired natural processes) and promoting education and recreation [59].

Located in southwestern Spain (Andalusia Autonomous Community), Doñana National Park is a natural reserve that covers 54,252 ha, with a socioeconomic influence area of 200,601.86 ha. Due to its privileged geographical location between two continents and its proximity to the meeting point of the Atlantic Ocean and the Mediterranean Sea, this nature reserve represents the confluence of different ecosystems (marshes, shallow streams, and sand dunes) that make it to be classified as a single natural landscape. It is considered as the largest nature reserve in Europe and is declared as a World Heritage Site by United Nations Educational, Scientific and Cultural Organization (UNESCO) [60].

Dromedaries or one-humped camels were presumably introduced in this emplacement from the Canary Islands in 1829 by the Marquis of Molina as beasts of burden and finally disappeared for poaching practices by the 1950s after having escaped from herds to the wild [61,62]. Fortunately, domestic dromedaries have been progressively reintroduced in limited herds in this reserve intended to serve as an ecotourism icon.

With an estimated average of 250,000 visitors per year (2017 Summary Report of National Parks Network, Ministry for Ecological Transition and Demographic Challenge, Government of Spain), a total distribution area greater than other national parks where Canarian camels are raised (Teide National Park 18,990 ha; Timanfaya National Park 5107.50 ha) (Figure 1), the historical presence of these animals and its orographic and climatic characteristics, the Doñana National Park could become a potential location for camel breeding and thus contributing to the conservation of this threatened indigenous breed. What’s more, Andalusia is one of the main tourist destinations in Spain (Tourism Statistics, National Statistics Institute), which may be a potential issue when appealing to customers to be involved in this entertaining recreation.

The knowledge of the sociocultural dimensions of this tourism alternative will allow, on the one hand, its value enhancement in Doñana National Park, the strengthening of the socioeconomic status in the area of socioeconomic influence of the park’s natural and cultural values, and its constitution as a potential alternative economic niche for the conservation of Canarian camels. In the Canary Islands, where these activities are already part of the usual inventory of tourism companies and the camel is an identifying emblem within local sand landscapes, in addition to the aforementioned benefits, the results of this research will greatly facilitate the extent promotion and empowerment of these activities as a sustainable iconic tourist choice.

By conducting an on-site visitor questionnaire survey after the completion of the trip, we could assess visitors’ profiles, their general satisfaction with these activities, and their return intention probability. The questionnaire was available in four languages (Spanish, English, French, and Portuguese), and it was designed through a deep review of consumer behavior researches in domestic animal tourism.

These written questionnaires were completed from 1 July to 30 September 2019, which is high season for southern Spain and thus, the business has higher demand. All the camel-riding consumers during the research period were asked to voluntarily fulfil the questionnaire. They all received basic training and guidance on the best practices for riding camels before the walk started.

The visitor survey was designed to collect multiple data. First of all, demographics (gender, age, and nationality), sociotemporal variables (month of visit and travel companions), other personal data of interest (study level, previous experience in camel riding, and personal perception of involved camels’ well-being), and factors affecting the decision-making process in tourism customers when choosing a destination (interests, motivations, and business duties). Riders’ overall satisfaction was assessed through the evaluation of staff performance, camel behavior/welfare, and service quality. Last, customer knowledge about camel functionalities, previous experience in camel-riding, and social awareness/agreement with the potential of such touristic rides in endangered breeds conservation and intention to return were considered. Staff performance appraisal includes language abilities, knowledge of camels and nature, manners, social abilities, and willingness to serve [63]. Service quality was evaluated through the dynamic and varied nature of the ride, consumer perception on whether walk length is proper or not, and overall perceived quality. Relationship between price and quality-quantity can be inferred from global customer contentment on just mentioned characteristics (short, adequate, or long journey, and whether the user suggests some improvement or not), based on the relative importance of price in quality and value judgments for consumer’s satisfaction and return intention probability [64]. For instance, Homburg et al., [65] reported that as satisfaction increases, the negative impact of the magnitude of a price increase is weakened.

According to literature, there is no single type of Likert scale that fits best to all the issues [66]. For this reason, we used a mixed-scale approach to measure customer satisfaction and attitudes towards camel rides. The number of items within each scale was also variable depending on the characteristic being evaluated. Travel motivation, staff performance, and quality service were asked to be measured on a 10-point Likert scale. On the other hand, camel behavior and welfare as well as personal attitudes towards camel tourism and other functional niches were rated on a 5-point Likert scale. When evaluating services, atmosphere and expectations fulfilment in tourism, longer response scales are preferable since they will reflect reliable distinctions among individuals’ responses and thus increase total valid score variance, measurement precision, and method validity [67,68]. Contrastingly, for psychological constructs (i.e., personality traits and behavioral intentions or attitudes towards a certain topic), rating scales are preferred to be odd-numbered as they include a mid, neutral alternative that allows respondents to express a neutral opinion and minimize potential ambiguity between adjacent categories [66], which should preferably be avoided when evaluating behavior.

### 2.3. Data Statistical Analysis

#### 2.3.1. General Model: Factors and Variables Considered

The questions included in the on-site visitor questionnaire survey were turn into the independent factors and dependent variables considered for the general model used in this study. Factors considered in the general model were organized in clusters, namely, customer and trip profile, decision-making motivating factors, staff performance, camel behavior, quality of riding route, previous experience, and customer impressions, whereas dependent variables were considered in the cluster of customer satisfaction and loyalty. Independent factors and dependent variables comprising each cluster are shown in Table 2.

#### 2.3.2. Parametric Assumptions Testing and Approach Decision

Shapiro-Francia W’ test (for 5 ≤ n ≤ 1000 samples) was performed to study data distribution using the Shapiro-Francia normality routine of the Stata Version 15.0 software [69]. Levene’s test was performed to determine the homogeneity of variance across groups using the explore procedure of the descriptive statistics package in SPSS Statistics, Version 25.0, IBM Corp. [70]. As provided parametric assumptions were grossly violated (Appendix A) and the strong subdivision occurring when testing for the effect of multiple factors may condition results, a Bayesian approach was followed to avoid power and precision loss in a relatively small data sample context as suggested by Hox et al. [71] and Lee and Song [72].

#### 2.3.3. Relationship between Factors and Variables

Once parametric assumptions had been tested and as a previous step to simple linear regression, potential multicollinearity and factor redundancy problems were discarded using Bayesian inference for Pearson correlations. Bayesian inference for Pearson correlations was performed to identify the relationship between variable pairs to detect and discard a multicollinearity issue between redundant factors (generalized Pearson correlation coefficient over 0.8 across variable pairs [73]) using the Pearson correlation task from the Bayesian statistics procedure in SPSS Statistics, Version 25.0, IBM Corp. [70].

#### 2.3.4. Multifactorial Dimensionality Reduction

Multifactorial dimensionality reduction can help to identify meaningful capturing the information (variability) for specific sets of factors within a dataset. One of the reference methods to perform dimensionality reduction effectively is principal component analysis (PCA). Kaiser-Meyer-Olkin (KMO) test of sampling adequacy and Bartlett’s test of sphericity was computed to establish the validity of the data set. Bartlett’s test of sphericity was performed to determine whether variables are unrelated and therefore unsuitable for structure detection. Initial and extraction communalities were assessed to determine which variables should be maintained or discarded from PCA. The cumulative proportion of variance criterion was finally employed to determine the number of components to extract. Cronbach alpha statistic was used to determine the reliability and validity of the reduced variable set. All statistical tests referred above were performed using SPSS Statistics for Windows statistical software, Version 25.0, IBM Corp. [70]. Component loadings lower than |0.5| were ruled out provided their confounding nature. Varimax with Kaiser normalization rotated component was performed to reduce the number of factors to be included in our general model [74], as it identifies which of all the factors considered are able to capture the maximum fraction possible of the variability in the dataset (significantly loaded factors ≥ 0.5).

#### 2.3.5. Bayesian Linear Regression Modelling for Customer General Satisfaction and Return Intention Probability

Once meaningful variability capturing factors had been identified (significantly loaded factors for each of the six dimensions identified by PCA), we built two separate simple linear regression predictive models comprising those factors holding a significant linear relationship with dependent variables of customer general satisfaction and return intention probability in regards to camel tourist walks (dependent variables), respectively. To this aim, the Linear Regression package from the Bayesian statistics task of SPSS Statistics for Windows, Version 25.0, IBM Corp. [70] was used. A full description of the algorithms used by SPSS to perform Bayesian Inference on Multiple Linear Regression Models in this study can be found in the public document IBM SPSS Statistics Algorithms Version 25.0. by IBM Corp. [70]. The Bayesian Linear Regression test routine of the Linear Regression and related package of the Stata Version 16.0 software process was used to compute posterior distribution statistics for each factor included in each model for each dependent variable.

#### 2.3.6. Model Validity

Acceptance rate, efficiency, and Monte Carlo standard error (MCSE) were used to determine the validity of the Bayesian methods implemented. The acceptance rate is the proportion of proposed values of *β* that were included in our final Markov chain Monte Carlo (MCMC) sample. The asymptotically optimal acceptance rate is 0.234 under quite general conditions [75]. Efficiency describes mixing properties of the Markov chain. High efficiency means good mixing (low autocorrelation) in the MCMC sample, and low efficiency means bad mixing (high autocorrelation) in the MCMC sample. An efficient Metropolis–Hastings (MH) sampler has an AR between 15% and 50% [76] and low autocorrelation and thus relatively large effective sample size (ESS) for all model parameters. The Monte Carlo standard error (MCSE) is an approximation of the error in estimating the true posterior mean.

Bayesian statistics predictive accuracy of the model [77] can be estimated through posterior predictive checking [78]. The posterior predictive *p* values (PPP values), as defined in Appendix C of Lee and Song [79], were computed as a goodness of fit measure for the model being tested, with a value around 0.5 indicating a plausible good-fitting model and values toward the extremes of 0 or 1 indicating that the model is not plausible. This is because the PPP values is the proportion of time during an MCMC run that a chosen test statistic, generated from a distribution predicted by the model, is higher than the test statistic generated from the distribution of the actual input data.

The marginal likelihood, also known as the evidence, or model evidence, is the denominator of the Bayes equation. Log marginal likelihood was performed to determine the performance of data fit implementing a penalty as model complexity increases [80]. The model reporting the highest log marginal likelihood is precisely the model that is the best sequential predictor of the data tested according to the log scoring rule [81]. A difference of 0.01 between two log-likelihood values is considered to be the same model. A difference of more than 3 log likelihood units (considered as “strong evidence against competing model”) can be used as threshold for accepting a more parameter-rich model [80]. A positive log likelihood means that the likelihood is larger than 1. This is possible because the log likelihood is not itself the probability of observing the data, but just proportional to it [82].

The ability to explain or capture the variability observed in the data set being studied (AIC and AICc) and the predictive potential (BIC) of the model designed for the data being modeled were computed as well [83,84,85]. In statistics, the Bayesian information criterion (BIC), Schwarz information criterion (SIC), Schwarz Bayesian criterion (SBC), or Schwarz Bayesian information criterion (SBIC) is a criterion for model selection among a finite set of models; the model with the lowest BIC is preferred. BIC is defined as—2 times the approximated log marginal likelihood (multiplied by—2 for historical reasons given most goodness-of-fit statistics are on the deviance scale). It is calculated as—2*L_m_* + *m*ln*n*, where *n* is the sample size, *L_m_* is the maximized log-likelihood of the model, and *m* is the number of parameters in the model. The index takes into account both the statistical goodness of fit and the number of parameters that have to be estimated to achieve this particular degree of fit, by imposing a penalty for increasing the number of parameters [83,84]. Using BIC, we can approximate the Bayes factor between two models by their R-squared and the numbers of predictors used in the models, when we have large sample of data, which makes the comparison more effective. Additionally, mean square error was computed to measure how close a regression line is to a set of points, that is how good a certain model fits the data being observed and to calculate the minimum mean-square error (MMSE).

## 3. Results

### 3.1. Respondents Sample

The total number of respondents was 131 (55 males and 76 females), that is the total number of camel-riding consumers during the research period (1 July to 30 September 2019). Customer qualitative profile (age, nationality, study level, and travel companions) is represented in Figure 2. Table 1 compiles quantitative demographics of camelback riding customers.

### 3.2. Descriptive Statistic Analysis

A summary of the descriptive statistics for the variables and potential conditioning factors of customer general satisfaction and return intention probability in regards to camel tourist walks is included in Table 2. In regards to user complacency on walk overall quality, 62% of respondents were satisfied and highly satisfied.

### 3.3. Dimensionality Reduction

Table 3 and Table 4 show a summary of general model clusters and the factors that each of them comprised, Varimax with Kaiser normalization rotated component loadings for each of these factors, eigenvalues, and percentage of variability in the dataset that is explained by each resulting principal component dimension.

Considering the factors of the general model clustered together on each principal component dimension, we determine the existence of the following factor dimensions: Principal Component Dimension 1 (PC1) or staff performance and trust-based camel-human partnership, Principal Component Dimension 2 (PC2) or sociospatial motivating factors, Principal Component Dimension 3 (PC3) or diverse and humane close interaction, Principal Component Dimension 4 (PC4) or camel behavior and performance, Principal Component Dimension 5 (PC5) or sociotemporal context, and Principal Component Dimension 6 (PC6) or camelback riding previous experience.

Varimax with Kaiser normalization rotated component loadings determined Principal Component Dimension 1 (PC1) or staff performance and trust-based camel-human partnership comprised the factors of language abilities, staff’s camel knowledge, nature knowledge, manners, social skills, and willingness to serve; Principal Component Dimension 2 (PC2) or socio-spatial motivating factors comprised the factors of customers’ camel knowledge, environmental knowledge, Andalusian culture, Andalusian friends/relatives, special event in Andalusia, conference/meeting, education/research, or business; Principal Component Dimension 3 (PC3) or diverse and humane close interaction comprised the factors of route being varied and adequately long, personal impression on involved camels’ welfare, and familiarity towards worldwide camel uses; Principal Component Dimension 4 (PC4) or camel behavior and performance involved the factors of camel being unfocused/distracted, fearful, surprised or rejective, and the knowledge of customers about functional niches of camels worldwide; Principal Component Dimension 5 (PC5) or sociotemporal context comprised the factors of month of the visit, animal involved in the touristic rides, travel companion, and the knowledge of customers about functional niches of camels worldwide; Principal Component Dimension 6 (PC6) or camelback riding previous experience which was constituted by previous experience and time to previous experience of camel rides.

### 3.4. Relationship between Factors and Variables

Appendix A summarizes the estimated sample Pearson correlation coefficient and the Bayes factors. For almost all variable pairs, the estimated Pearson correlation coefficient reaches values from −0.46 to 0.91, and the corresponding Bayes factor was always > 1. These values indicate that the factors and variables considered in this study provide significant evidence in favor of a moderate to strong linear correlation.

### 3.5. Reduced Linear Models: Bayesian Linear Regression Modelling for Customer General Satisfaction and Return Intention Probability

Determination coefficient [86] or percentage of variance captured and significance for each of the models comprising the variables significantly loading on each of the six principal component dimensions are provided in Table 4. Table 4 also suggests model comprising significantly loading factors in dimension 3 and model comprising significantly loading factors in dimension 6, which significantly explain and predict for customer general satisfaction and return intention probability, respectively, than others comprising just the intercept. The rest of the dimensions did not report a significant result for any of the variable clusters previously determined by principal component analysis, revealing the relationship with them, and the dependent variables may be other than linear, hence a direct predictive simple relationship cannot be determined, then were not considered in reduced linear models for customer general satisfaction and return intention probability.

Predictors in model comprising variables in dimension 3 were able to capture 43.3% of the variability of customer general satisfaction. Contrastingly, predictors in the model comprising variables in dimension 6 were able to capture 28.7% of the variability in return intention probability. The 95% credibility interval shows that there is a 95% probability that these regression coefficients in the population lie within the corresponding intervals. When 0 is not contained in the credibility interval, a significant effect for such factor is detected.

### 3.6. Model Validity

The posterior predictive *p* values (PPP values) for both models were around 0.5. This value indicates moderately plausible good-fitting models. Similarly, the difference of one log-likelihood unit and BIC can be considered as evidence of model for return intention probability better predictive ability. The summary of the results for the parameters of validity of both models is reported in Appendix A. When R^2^, AIC, AICc, and BIC criterion were compared between models, the model for customer general satisfaction presented a slight ability to capture variability as suggested by R^2^ values.

## 4. Discussion

### 4.1. Theoretical and Managerial Implications

Camel-riding experiential consumption in southern Spain was evaluated through the identification of decision-making, customer general satisfaction and return intention probability potentially influencing factors. This animal-close experience is an outdoor exotic adventure that includes physical challenging, cultural exchange, and activities in nature, with relatively high levels of sensory stimulation [87]. Tourists parties (mainly adult relatives and friends) from foreign countries and national regions where camels are not traditionally reared nor its functional niches are well known (Figure 2 and Table 1), tempted to visit faraway destinations for different reasons (cultural and natural heritage, leisure iconic activities, and/or professional/business goals) [88], are finally involved in such activities in these places where camels are reared and act as a potential tourism attraction (PC2, Table 3).

This finding addresses the fact that camelback riding may not be a major vacation preference to consumers when planning their holidays and tour agencies may not include this entertaining recreation in their tour guides and proposals; as supported by Figure 2, with only one tour group being sampled. Contextually, tourism operators and stakeholders have a central role when revalorizing camel rides to make them become an adjacent constructive element within the cultural heritage that makeup the local tourist space within a sustainable framework.

With a mild climate and a wide variety of sceneries, southern Spain is one of the most popular places as a tourist destination in this country both for national and international citizens. These circumstances, strongly linked with the fact that camels are progressively being raised in this region for its oroclimatic peculiarities (sandy landscapes of southwestern areas), constitute a huge opportunity for their functional revalorization as leisure animals lastly aiming the conservation of the autochthonous endangered Canarian camel, the only genetic resource of such nature in Spain and Europe [89].

As a key part of this conservation strategy, the enhancement of wide-audience environmental education programs financially supported by competent authorities and which contemplates camels in its academical content could not only help to enrich consumer profile by generating interests towards these animals in the youngers (Figure 2) but also help tourism enterprises to experience higher demand for such type of entertainment. Within this initiative, individuals directly involved in this sector and with proven experience in the daily management of these animals should be the leaders. Therefore, interested users will be provided with empirical knowledge and skills concerning camel handling prior to the walk and be trained to manage any incidence that may occur during the journey, intending to make it as enjoyable as possible. In fact, staff performance (general attitudes and derived service quality) and trust-based animal-human relationships (fearless, focused, and trusted camel) are service characteristics that affect customer satisfaction, according to our results (PC1, Table 3). Likewise, a mistrustful camel may be the result of a poor training workout, hence an indirect, negative judgment of previous staff performance and fair estimate of user satisfaction itself [90].

The intrinsic characteristics of the walk most valued by riders are its diversity, their perception towards the adequate length of the ride, and overall quality (PC3, Table 3). Since user complacency was high in this regard (62% of respondents were satisfied and highly satisfied), longer excursions could be a better option for the camel and provide further emotional support for riders than shorter jaunts only if taking prolonged breaks on multiple occasions throughout the journey, generally tied in with hydrating the camel, even though guides are quick to ensure guests that the camels can last great lengths without water for its particular physiological adaptations [91]. Instead, the indirect estimation of respondents’ perception about pricing shows this factor is not as relevant as lengthiness sensory stimulation during the journey when explaining customer satisfaction. Zeithaml [64] proposes that price functions as a surrogate for quality when no more intrinsic cues are available, which is not applicable in this particular case.

In the same dimension (PC3), the well-being of the camels involved according to subjective criteria and the consumer’s familiarization with camel functional niches (in our case, the vast majority (74%) were not at all or not very familiar with this item) influenced and predicted for the general perception and satisfaction (PC3, Table 3 and Table 4) [1]. Our results suggest that the more familiarized users are with camel functionalities, the more criticism they develop when evaluating the impact of these activities on camel welfare and the fulfillment of personal expectations. To avoid falling into unfounded evaluations as well as to provide unfamiliarized consumers with objective information with a proven scientific basis in this regard, herdsmen and animal welfare legislative authorities are responsible for the elaboration and dissemination of codes of good practice for these activities, promoting adequate training sessions for camels, and encouraging consumers’ coaching prior to carrying out the activity.

Although some breeders might abuse or neglect camels for their conception as trade tools for their own benefit, most of the animal owners fortunately realize that well-reared and healthy animals are the best working ones [92]. Here, customers are fundamental when judging owners’ attitudes and handling practices towards the camels to decide whether or not it is convenient to ride the camel/s and recommend the same trekking company to other travelers looking for a cruelty-free camel-riding experience. Ultimately, the overall performance of the activity will be satisfactory, both in terms of impact on animal welfare and overall consumer satisfaction [56].

In the same frame-analytic scenario, the variables integrating the fourth dimension (PC4, Table 3) might reinforce the abovementioned customer facet. When camel functional niches are well known for consumers, these may be more likely to judge camels for behavioral features that involve mistrust, fear, or rejection as they are factors with derogatory effects on the performance and quality of the route and, therefore, consumer satisfaction.

These undesired behavior traits not only could be derived from poor habit training and idiosyncratic components but also affect camel performance by overstraining practices and improper-fitting saddles. Given that these activities are concentrated in a short period of time and the number of animals involved is limited, since it is an endangered species and, in the particular case of Doñana, in process of functional reintroduction, the animals may be susceptible to cumulative fatigue and tiredness as the business takes advantage of this high season as much as it can because it is the most economically fruitful time for their self-financing and maintenance for the rest of the year. Besides, due to the fact that is an activity preferable to experience accompanied, it is most common for camels to support two riders on its back (PC5, Table 3). However, like those that offer camel riding rely on tourists to make money and finally supply what the tourists demand, they must be conscious about increasing global concerns on animal welfare because unhappy tourists that feel the animals are being treated badly or not getting enough rest will not be engaged in these activities and influence enforcement opportunities, by reporting poor practice when it occurs.

In this sense, considering that Andalusia continues to be a tourist preference and the Doñana National Park receives visitors during the rest of the year, this overburdening fatigue can be avoided by proper fairly constant training and physical stimulation as long as camel rides and camel-close interactions could become an alternative for other recreational experiences with transversal education purposes and serve as a therapeutic resource for social groups with special needs [93]. According to this statement, Majchrzak et al. [94] maintained that these activities are not a stressful experience for animals and could be a form of environmental enrichment based on cortisol levels in saliva. Such evidence provides a further criterion to promote these activities and thus, enhance their potential to become a sustainable business supporting camel farms’ long-term viability, regional economic growth, and natural resources conservation, by not standing for unethical practices and having a policy on tourism to be able to justify the options they offer.

Last, the combination or co-occurrence of all considered sociocultural dimensions of camel rides in this pioneer research approach would not only affect satisfaction but also return intention probability. With large added value, positive previous experiences seemed to be a conditioning and predicting factor for tourists when seeking for further involvement in this emerging exotic tourism attraction (PC6, Table 3 and Table 4).

### 4.2. Limitations and Future Research Approaches

Despite this study offers an all-round and overall view of camelback riding as emerging tourist experiences with a large potential to contribute to sustainable socioeconomic development and natural resources conservation, we discuss possibilities for future research within this academic field. First, despite of the fact that pricing is not presumably influencing customer satisfaction and return intention probability, the categorical inclusion of users’ profession would help to improve the knowledge of the socioeconomic level of camel riding consumers. In addition, we will be able to test whether a new experiential dimension, including variables already considered (sex, age, country of origin, and study level) and the new one proposed, appears on the scene. Income levels could be a potential factor conditioning users’ expenditure in destinations where camel rides are offered, in such a manner that tourists cannot spend more money than which they have already spent in secondary activities, as camelback tourist rides actually are. Further guidance would be provided for stakeholders when aiming to implement strategic actions (i.e., competitive but affordable prices).

Within the same framework, it would also be interesting to investigate the different sources of information from which consumers had learned about the places where camel rides were available as tourism attractions and whether prior to booking they had knowledge about the welfare and general care of the animals in a specific location. Findings will enable us to identify weak points in the value chain of this touristic alternative and propose multivariate sources to meet its specific needs and goals as well as distribute the latest product information.

A third consideration is the potential utility of an ethological and biomechanical characterization of the camel breed. The first one is a useful tool to empirically evaluate the real impact of tourist rides in camel welfare and general health status by comparing behavioral features that camels show during rides and natural behavioral patterns. In this regard, investigate if ridden camel behavior whether is sexually dimorphic could also provide valuable information. Regarding kinetics assessment, it allows to correlate camel physical performance and constitution with customer satisfaction, as camels with greater lightness and agility are expected to put up with a strict sports practice and reach the desired performance during riding walks. This functional appraisal [21,26,27] and the results of the customer satisfaction surveys will be traced for each animal to identify which physical attributes and personality traits would be the most appropriate for pleasant customer experience. Age must be an influencing factor to being considered in this regard so as to ultimately estimate camel’s useful life to remain in service for this purpose. Therefore, new selection criteria will be embraced aiming for the genetic improvement and conservation of Canarian camel, mostly relegated to tourism and leisure activities.

Different types of camel saddles would also be a good opportunity to inference consumer perception on the easiness and comfortability of camel riding and managerial implications on camel well-being.

Last, the replication of this research proposal in other geographical locations where Canarian camels are present, such as the Canary Islands, may give greater validity to our findings and make results more likely to be generally applied to the whole endangered population.

## 5. Conclusions

This paper presents a pioneer systematic approach of the sociocultural dimensions that surround camel tourism, particularly camelback riding tours. This exotic leisure activity attracts flocks of tourists seeking for traversing sandy environments on the back of these animals in caravans while supporting their positive welfare at the time of encountering. Essentially, the diverse social and cultural determinants that potentially contribute to creating this experience can be summarized as follows: staff performance, culture geography, diverse and humane close interaction, camel behavior and performance, sociotemporal context, and positive previous experience. The resultant specific-consumer knowledge assessment not only supplies the dire need for further practical research of animal-based tourist experiences but also capacitates involved stakeholders to refine their abilities to successfully meet and satisfy customer’s expectations according to the current social concerns, travel decision-making criteria, demands, and sustainability issues within this adventure enterprise. Tourism companies will, in turn, gain customer loyalty and fortify revenue opportunities. Contextually, this empirical marketing segmentation sheds light on the potential functional reintroduction of Canarian camels in national-protected areas out-of-the-archipelago that reunite geological and orographic conditions for these species’ survival, thus contributing to the long-term viability of this autochthonous endangered breed and avoid its extinction.

## Figures and Tables

**Figure 1 animals-10-01703-f001:**
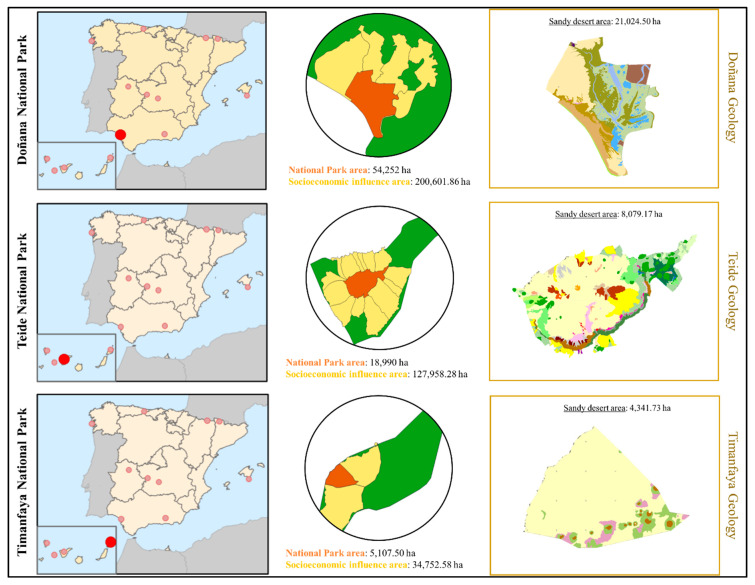
Quantum Geographical Information System (QGIS) maps displaying geographical location, surface, and geology of Doñana, Teide, and Timanfaya National Parks. Sandy desert area’s extension within each National Park is light-yellow colored.

**Figure 2 animals-10-01703-f002:**
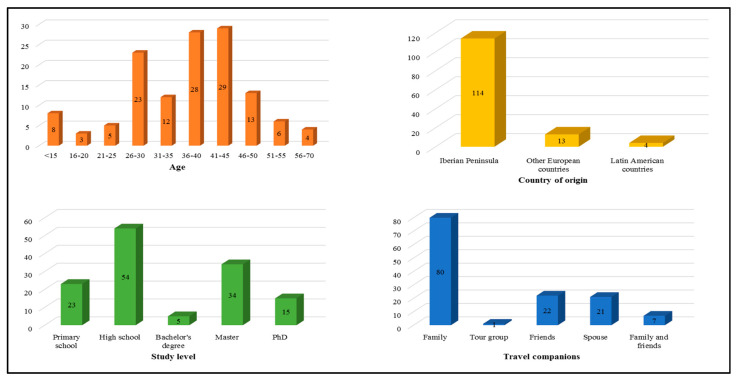
Outputs from camelback riding customer’s qualitative profile frequency analysis.

**Table 1 animals-10-01703-t001:** Camelback riding customer’s demographics.

Country of Origin		Number of Visitors
Iberian Peninsula	Northern provinces	33
	Middle provinces	34
	Southern provinces	46
	Balearic Islands	1
Other European countries	Denmark	3
	France	2
	Germany	5
	Italy	3
Latin American countries	Argentina	1
	Cuba	1
	Honduras	1
	Venezuela	1

**Table 2 animals-10-01703-t002:** Summary of descriptive statistics of factors and variables of customer’s general satisfaction and return intention probability in regards to camel tourist walks.

Clusters	General Model Factors and Dependent Variables	Mean	SEM	Median	Mode	SD	CV	Variance	IQR	Min	Max	Percentile 25	Percentile 75
Customer and trip profile	Month of visit	8.33	0.12	8.00	8.00	1.33	15.93	1.76	8.00	2.00	10.00	8.00	9.00
	Sex	1.59	0.04	2.00	2.00	0.49	31.07	0.25	1.00	1.00	2.00	1.00	2.00
	Age	5.73	0.19	6.00	7.00	2.12	37.05	4.51	9.00	1.00	10.00	4.00	7.00
	Country origin	1.15	0.04	1.00	1.00	0.44	37.83	0.19	2.00	1.00	3.00	1.00	1.00
	Study level	3.73	0.12	3.00	3.00	1.38	37.02	1.91	6.00	1.00	7.00	3.00	5.00
	Travel companion	2.05	0.12	1.00	1.00	1.40	68.44	1.97	5.00	1.00	6.00	1.00	3.00
Decision-making motivating factors	Camel knowledge	7.09	0.32	8.00	10.00	3.18	44.78	10.08	9.00	1.00	10.00	5.00	10.00
	Environmental knowledge	7.62	0.25	8.00	10.00	2.40	31.51	5.77	9.00	1.00	10.00	5.75	10.00
	Andalusian culture	7.28	0.27	8.00	8.00	2.44	33.50	5.95	9.00	1.00	10.00	6.00	9.00
	Andalusian friends/relatives	6.46	0.45	8.00	10.00	3.74	57.85	13.96	9.00	1.00	10.00	1.75	10.00
	Special event in Andalusia	5.49	0.51	6.00	1.00	3.58	65.26	12.84	9.00	1.00	10.00	1.00	9.00
	Conference/meeting	3.87	0.60	1.00	1.00	3.74	96.54	13.96	9.00	1.00	10.00	1.00	8.00
	Education/research	4.76	0.52	5.00	1.00	3.54	74.39	12.54	9.00	1.00	10.00	1.00	8.00
	Business	3.14	0.53	1.00	1.00	3.17	101.05	10.07	9.00	1.00	10.00	1.00	5.00
	Holidays	9.41	0.17	10.00	10.00	1.82	19.30	3.30	9.00	1.00	10.00	10.00	10.00
Staff performance	Language abilities	9.48	0.08	10.00	10.00	0.94	9.96	0.89	5.00	5.00	10.00	9.00	10.00
	Camel knowledge	9.75	0.07	10.00	10.00	0.74	7.59	0.55	6.00	4.00	10.00	10.00	10.00
	Nature knowledge	9.49	0.10	10.00	10.00	1.15	12.09	1.32	7.00	3.00	10.00	9.00	10.00
	Manners	9.80	0.05	10.00	10.00	0.61	6.19	0.37	4.00	6.00	10.00	10.00	10.00
	Social skills	9.70	0.08	10.00	10.00	0.91	9.36	0.83	8.00	2.00	10.00	10.00	10.00
	Willingness to serve	9.77	0.08	10.00	10.00	0.92	9.39	0.84	9.00	1.00	10.00	10.00	10.00
Camel behavior	Unfocused/distracted	1.46	0.09	1.00	1.00	0.86	58.90	0.74	4.00	1.00	5.00	1.00	2.00
	Calm/awaiting	4.24	0.11	5.00	5.00	1.25	29.53	1.57	4.00	1.00	5.00	4.00	5.00
	Mistrustful	1.31	0.09	1.00	1.00	0.79	60.23	0.62	4.00	1.00	5.00	1.00	1.00
	Fearful	1.22	0.08	1.00	1.00	0.67	55.16	0.45	4.00	1.00	5.00	1.00	1.00
	Depressed	1.12	0.06	1.00	1.00	0.54	47.95	0.29	4.00	1.00	5.00	1.00	1.00
	Curious	2.93	0.14	3.00	3.00	1.25	42.76	1.57	4.00	1.00	5.00	2.00	4.00
	Surprised	1.52	0.10	1.00	1.00	0.84	55.07	0.70	4.00	1.00	5.00	1.00	2.00
	Rejection	1.11	0.04	1.00	1.00	0.39	34.77	0.15	2.00	1.00	3.00	1.00	1.00
	Indifferent/irresponsive	1.48	0.11	1.00	1.00	0.95	64.46	0.91	4.00	1.00	5.00	1.00	2.00
	Cautious	3.05	0.16	3.00	3.00	1.43	46.95	2.05	4.00	1.00	5.00	2.00	4.00
	Nervous	1.37	0.09	1.00	1.00	0.76	55.69	0.58	3.00	1.00	4.00	1.00	1.00
Quality of riding route	Secure	9.75	0.05	10.00	10.00	0.58	5.99	0.34	3.00	7.00	10.00	10.00	10.00
	Interesting	9.21	0.10	10.00	10.00	1.08	11.77	1.18	5.00	5.00	10.00	9.00	10.00
	Varied	8.53	0.15	9.00	10.00	1.57	18.39	2.46	8.00	2.00	10.00	7.25	10.00
	Appropriately long	8.87	0.17	10.00	10.00	1.77	20.00	3.15	9.00	1.00	10.00	8.00	10.00
	Walk overall quality	3.02	0.10	4.00	4.00	1.15	38.08	1.32	3.00	1.00	4.00	2.00	4.00
Previous experience	Previous experience	1.66	0.04	2.00	2.00	0.48	28.73	0.23	1.00	1.00	2.00	1.00	2.00
	When did previous experience take place?	9.62	1.07	9.50	10.00	6.91	71.83	47.75	26.00	1.00	27.00	3.75	15.00
	Comparison between experiences	4.05	0.18	4.00	5.00	1.11	27.36	1.23	4.00	1.00	5.00	3.00	5.00
	Did you receive previous training before the walk?	1.00	0.00	1.00	1.00	0.00	0.00	0.00	0.00	1.00	1.00	1.00	1.00
Customer impressions	Personal impression on involved camels’ welfare	2.50	0.05	3.00	3.00	0.59	23.48	0.35	2.00	1.00	3.00	2.00	3.00
	Do you think this tourism activity has wide impacts on camel health and welfare?	3.13	0.10	3.00	4.00	1.12	35.75	1.25	4.00	1.00	5.00	2.00	4.00
	Personal impression on camel riding as a sustainable tourism activity	4.45	0.07	5.00	5.00	0.77	17.24	0.59	4.00	1.00	5.00	4.00	5.00
	Easiness of camelback riding	3.02	0.08	3.00	3.00	0.92	30.40	0.84	3.00	1.00	4.00	3.00	4.00
	Comfortability of camelback riding	2.53	0.07	3.00	3.00	0.66	25.97	0.43	2.00	1.00	3.00	2.00	3.00
	Familiarity towards worldwide camel uses	2.15	0.11	2.00	1.00	1.06	49.16	1.12	4.00	1.00	5.00	1.00	3.00
	What do you think camels are raised for?	7.71	0.56	5.00	5.00	6.17	80.05	38.09	26.00	1.00	27.00	3.00	10.00
	Consciousness about the usefulness for endangered camels breeding and conservation	3.30	0.11	3.00	3.00	1.26	38.30	1.60	4.00	1.00	5.00	2.00	4.00
	Walk length	1.79	0.04	2.00	2.00	0.41	22.96	0.17	1.00	1.00	2.00	2.00	2.00
Customer satisfaction and loyalty	General satisfaction	2.36	0.06	2.00	2.00	0.67	28.43	0.45	4.00	1.00	5.00	2.00	3.00
	Return intention probability	3.98	0.09	4.00	4.00	0.98	24.60	0.96	4.00	1.00	5.00	4.00	5.00

**Table 3 animals-10-01703-t003:** Summary of general model clusters and factors that each cluster comprised and Varimax with Kaiser normalization rotated component loadings for each of these factors.

Clusters	General Model Factors	PC1	PC2	PC3	PC4	PC5	PC6
Customer and trip profile	Month of visit	−0.003	0.322	−0.200	0.433	0.670	−0.411
	Animal	−0.064	0.191	−0.193	0.464	0.796	−0.293
	Sex	−0.096	0.270	−0.106	−0.137	−0.026	0.070
	Age	−0.082	0.120	0.026	−0.336	−0.160	−0.211
	Country of origin	0.061	−0.339	0.076	0.319	−0.056	0.178
	Study level	0.043	−0.003	0.091	−0.335	−0.299	0.094
	Travel companion	0.017	−0.061	0.148	−0.414	−0.766	0.270
Decision-making motivating factors	Customers’ Camel knowledge	0.260	0.743	−0.104	−0.090	−0.007	0.113
	Environmental knowledge	0.050	0.767	−0.049	−0.142	−0.086	0.198
	Andalusian culture	0.009	0.808	−0.089	−0.192	−0.185	0.114
	Andalusian friends/relatives	−0.009	0.695	−0.034	−0.328	−0.077	0.116
	Special event in Andalusia	−0.002	0.789	−0.091	−0.169	−0.058	0.037
	Conference/meeting	0.020	0.865	−0.136	−0.009	−0.019	0.084
	Education/research	0.023	0.872	−0.108	0.019	−0.033	0.046
	Business	0.034	0.802	−0.114	0.227	−0.131	0.142
	Holidays	0.422	0.419	−0.127	−0.020	0.208	−0.188
Staff performance	Language abilities	1.171	−0.054	−0.067	−0.065	−0.032	−0.080
	Staff’s camel knowledge	1.172	−0.052	−0.060	−0.064	−0.026	−0.078
	Nature knowledge	0.881	−0.048	−0.034	0.026	−0.040	0.003
	Manners	1.171	−0.060	−0.069	−0.069	−0.032	−0.077
	Social skills	1.171	−0.061	−0.059	−0.066	−0.032	−0.079
	Willingness to serve	1.172	−0.061	−0.058	−0.063	−0.033	−0.079
Camel behavior	Unfocused/distracted	0.082	−0.081	−0.015	0.571	−0.406	−0.203
	Calm/awaiting	0.270	−0.097	0.338	0.047	0.049	−0.308
	Mistrustful	-1.168	0.067	0.068	0.066	0.029	0.075
	Fearful	0.016	0.106	−0.004	0.707	−0.500	−0.137
	Depressed	0.007	−0.091	0.032	0.454	−0.273	−0.094
	Curious	0.048	0.052	0.030	0.412	−0.068	−0.028
	Surprised	0.026	0.303	−0.048	0.694	−0.499	−0.112
	Rejection	0.025	0.177	−0.011	0.700	−0.499	−0.055
	Indifferent/irresponsive	−0.036	−0.304	0.048	0.134	−0.222	−0.139
	Cautious	0.058	0.220	−0.091	−0.135	0.287	0.189
	Nervous	−0.004	0.063	−0.025	0.393	−0.229	0.000
Quality of riding route	Secure	−0.018	0.278	0.275	−0.140	0.328	−0.176
	Interesting	0.430	0.172	0.406	0.143	0.196	0.131
	Varied	0.085	0.154	1.247	0.107	0.186	0.013
	Appropriately long	0.046	0.118	0.970	0.065	0.057	−0.031
Previous experience	Previous experience	0.225	−0.160	−0.196	0.423	0.358	0.843
	When did previous experience take place?	−0.189	0.175	0.138	−0.367	−0.313	−0.707
Customer impressions	Personal impression on involved camels’ welfare	0.102	0.235	0.663	0.027	0.009	0.268
	Personal impression on camel riding as a sustainable tourism activity	0.047	0.345	0.410	−0.008	−0.113	−0.003
	Easiness of camelback riding	0.067	0.149	0.469	0.086	−0.146	0.186
	Comfortability of camelback riding	0.224	−0.056	0.277	−0.001	−0.022	0.257
	Familiarity towards worldwide camel uses	0.039	0.260	−0.647	−0.084	0.035	0.164
	What do you think camels are raised for?	0.033	0.338	−0.056	0.671	−0.506	0.017
	Do you think this tourism activity has wide impacts on camel health and welfare?	0.131	0.374	−0.095	0.055	0.236	0.170
	Consciousness about the usefulness for endangered camels breeding and conservation	−0.169	0.428	0.151	−0.073	0.052	−0.153
	Walk length	−0.020	−0.081	0.314	0.099	0.341	0.064
Cronbach’s alpha (0.99)		0.873	0.845	0.741	0.779	0.744	0.705
Variability accounted for (based on total eigenvalues) (35.49%)		9.650	6.808	6.214	4.749	4.441	3.630

PC: Principal components dimensions. Grey shaded cells indicate significantly loaded factors (factor loading > |0.50|) for interpretation, hence variables clustered together per each principal component dimension.

**Table 4 animals-10-01703-t004:** Bayes factor model summary of model testing for customer general satisfaction and return intention probability in regards to camel tourist walks.

Parameters		PC1	PC2	PC3	PC4	PC5	PC6
Customer general satisfaction	Sum of squares	10.289	13.200	15.996	11.157	19.040	3.329
	df	19	27	20	35	40	5
	Mean square	0.542	0.489	0.800	0.319	0.476	0.666
	F	1.727	N/A	2.523	0.743	1.076	1.597
	Sig.	0.064	N/A	0.003	0.810	0.384	0.187
	R square	0.401	1.000	0.433	0.419	0.365	0.190
	Adj. R square	0.169	1.000	0.262	−0.145	0.026	0.071
	Bayes factor	0.000	N/A	0.200	0.000	0.000	0.017
Return intention probability	Sum of squares	12.166	22.000	29.862	42.923	45.291	11.418
	df	19	27	20	35	40	5
	Mean square	0.640	0.815	1.493	1.226	1.132	2.284
	F	0.701	N/A	1.645	1.470	1.331	2.738
	Sig.	0.800	N/A	0.067	0.125	0.142	0.035
	R square	0.210	1.000	0.326	0.582	0.412	0.287
	Adj. R square	−0.090	1.000	0.128	0.186	0.102	0.182
	Bayes factor	0.000	N/A	0.000	0.000	0.000	0.150

PC: Principal components dimensions. Grey shaded cells indicate significantly loaded factors (factor loading > |0.50|) for interpretation, hence variables clustered together per each principal component dimension.

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
