# Peer review of "Camel Genetic Resources Conservation through Tourism: A Key Sociocultural Approach of Camelback Leisure Riding"

_animals, 2020, doi:10.3390/ani10091703_

Round 1

Reviewer 1 Report

Authors should consider the following proposed changes:

  • Tables must be presented in Animals format, without intercell lines.
  • Why does Table 1 and its continuation present darkened cells. Please explain as a footnote of the table.
  • Certain sets of variables in Table 1 were not grouped into global categories, why? I would suggest grouping them as they seem to hold a similar background.
  • Line 103 – 104: Avoid overuse of the word purpose in correlative sentences.
  • Sample age range. Please confirm the minimum age of the animals involved in the touristic rides is 2 years old.
  • Many paragraphs comprise only three lines. Try to unite these small paragraphs in larger ones, which will improve the readability of the manuscript.
  • In the conclusion section. Be careful of the use of contractions such as What’s more, provided the informal character of such expression which would be inappropriate of a research paper.
  • The conclusion section should be written as a whole paragraph rather than smaller ones. Consider reducing the context to stick to the very conclusions that you draw after your results. Furthermore, conclusions should be obtained after your own results rather than speculating what could be after facts which were not dealt with in the paper.

Author Response

Reviewer 1

Comments and Suggestions for Authors

Authors should consider the following proposed changes:

  • Tables must be presented in Animals format, without intercell lines.

Resposne: Intercell lines were removed.

  • Why does Table 1 and its continuation present darkened cells. Please explain as a footnote of the table.

Response: Footnote was added.

  • Certain sets of variables in Table 1 were not grouped into global categories, why? I would suggest grouping them as they seem to hold a similar background.

Response: Variables were clustered together.

  • Line 103 – 104: Avoid overuse of the word purpose in correlative sentences.

Response: We changed repetitions of the word purposes by other synonyms.

  • Sample age range. Please confirm the minimum age of the animals involved in the touristic rides is 2 years old.

Response: We corrected age of the sample. Minimum age is 4 years old.

  • Many paragraphs comprise only three lines. Try to unite these small paragraphs in larger ones, which will improve the readability of the manuscript.

Response: Short paragraphs were unified.

  • In the conclusion section. Be careful of the use of contractions such as What’s more, provided the informal character of such expression which would be inappropriate of a research paper.

Response: Conclusion section was revised for colloquial expressions and these were changed to rather formal ones.

  • The conclusion section should be written as a whole paragraph rather than smaller ones. Consider reducing the context to stick to the very conclusions that you draw after your results. Furthermore, conclusions should be obtained after your own results rather than speculating what could be after facts which were not dealt with in the paper.

Response: The conclusion section was rewritten following the reviewer suggestion.

Reviewer 2 Report

The authors present a good explanation for the need of their research, and background surrounding camel tourism. However, the methodology used seems overly complicated and difficult to interpret. If the authors could give a better interpretation of their results, as well as integrate the summary statistics they have moved to an appendix, they could make a better case for their discussion section.  Detailed suggestions to the authors are listed below, but the main concern is a better explanation and interpretation of the methods used. This is accompanied with a lack of explanation and interpretation of the results which makes it hard for the reader to internalize the findings of the paper. I believe with additional details the contribution this paper provides may become clearer. Additionally, the authors should do a careful read for clarity and grammar. I have only pointed out a few editorial changes that should be made, but there are many more.

Introduction

It’s not clear what this is saying. Edit for clarity “This travelers’ engagement with a country or region's culture enhances the potential of smaller cultural and heritage attractions to come out into business marketing all over the world [5], but also force living local communities to experiment substantial cultural and social changes, both positive and negative.”

Methods

 “The total number of respondents was 131 (55 males, 76 females), that is the total number of 205 camel-riding consumers during the research period (1st July to 30th September 2019). Customer qualitative profile (age, nationality, study level and companions) is represented in Figure 1.” Please move this figure and it’s written description to the results section.

“For this reason, we used a mixed-scale model to measure customer satisfaction and attitudes towards camel rides.” The use of the word model here is a bit confusing as I was expecting an econometric model. I think a more appropriate term would be approach.

“Shapiro-Francia W' test (for 5≤n≤1000 samples) was performed to study data distribution using the Shapiro-Francia normality routine of the Stata Version 15.0 software.” I believe this is missing a citation

“As provided parametric assumptions were grossly violated (Supplementary Table S1) and the strong subdivision occurring when testing for the effect of multiple factors may condition results, a Bayesian approach was followed to avoid power and precision loss in a relatively small data sample context.” This also needs a citation

“Varimax with Kaiser Normalization Rotated Component was performed to reduce the number of factors to be included in our model.” Missing a citation

It is unclear to me how the Varimax with Kaiser Normalization Rotated Component is working with 2 different dependent variables customer general satisfaction and return intention probability. I think better interpretation of your findings would be helpful.

In general- it would be helpful to the reader to introduce the main model first, and then explain the corrections done. I was unclear what the end-goal was for this method until I reached the section on OLS.

“Using significantly loaded factors for each of the six dimensions identified by PCA, we built simple linear regression models to predict for customer general satisfaction and return intention probability in regards to camel tourist walks using the Linear Regression package from the Bayesian statistics task of SPSS Statistics for Windows, Version 25.0, IBM Corp.” Please write out the model

“The posterior predictive p values provide (PPP values) were computed as a goodness of fit measure for the model being tested.” It is unclear at this point what the model is?

Results

“A summary of the descriptive statistics for the variables and potential conditioning factors of customer general satisfaction and return intention probability in regards to camel tourist walks, is included in Supplementary Table S2.” Space permitting- I believe this is an important contribution and should be included in the main paper.

Table 1: The decision making factors- I am unsure of what it says as it is cut off- should be introduced previously in the methods. Also what do the grey boxes indicate? How are the numbers in the PC columns interpreted?

Table 1- “Considering the variables clustered together on each principal component dimension” can you give an example of this. For example the clustered variables X, Y, Z led us to determine dimension 1 etc.

“Table 2 also suggests model comprising significantly loading factors in dimension 3 and model comprising significantly loading factors in dimension 6 were more likely to explain and predict for customer general satisfaction and return intention probability, respectively, than others comprising just the intercept.” It is difficult to interpret what is going on here without having the models written out. I believed you were doing OLS, but we never see coefficients that will help you make the connection between what is important to the visitors and the dependent variable with any kind of magnitude scale.

“The posterior predictive p values for both models were around 0.5. This value indicates 396 moderately plausible good-fitting models.” It’s unclear what this is saying, and it needs a citation.

Discussion

Table 3 belongs in the results section

“In fact, staff performance and trust-based animal-human relationships are potential features affecting customer satisfaction, according to our results (PC1, Table 1). Likewise, a mistrustful camel may be the result of a poor training workout, hence an indirect, negative judgment of previous staff performance and fair estimate of user satisfaction itself [53].” Potential features? Did the results show it had an impact or not?

“Since user complacency was high in this regard (62 % of respondents were satisfied and highly satisfied)” this would be useful information to present in the results.

“Instead, the indirect estimation of respondents’ perception about pricing shows this factor is not as relevant as lengthiness sensory stimulation during the journey when explaining customer satisfaction.” I don’t see this in the results.

Minor/editorial:

Pg 1 line 29- edit for clarity and length “Camels are iconic and exotic regional pull elements within adventure travel companies 29 that promote eco-tourism activities with strong potential to generate revenues for local livelihoods and promote educational values towards nature and its conservation.”

Pg 1 line 25- delete may “satisfaction and loyalty in this tourism segment and that may made it socially differentiate”

“Bartlett's test of sphericity tests was performed to determine whether variables are unrelated and therefore unsuitable for structure detection.” Change to: Bartlett's test of sphericity was performed to determine whether variables are unrelated and therefore unsuitable for structure detection.

“Initial and extraction communalities were assessed to determine which variables should be maintained of discarded from PCA.” I believe of should be changed to or

Author Response

Reviewer 2

Comments and Suggestions for Authors

The authors present a good explanation for the need of their research, and background surrounding camel tourism. However, the methodology used seems overly complicated and difficult to interpret. If the authors could give a better interpretation of their results, as well as integrate the summary statistics they have moved to an appendix, they could make a better case for their discussion section.  Detailed suggestions to the authors are listed below, but the main concern is a better explanation and interpretation of the methods used. This is accompanied with a lack of explanation and interpretation of the results which makes it hard for the reader to internalize the findings of the paper. I believe with additional details the contribution this paper provides may become clearer. Additionally, the authors should do a careful read for clarity and grammar. I have only pointed out a few editorial changes that should be made, but there are many more.

Response: We thank the reviewer for his/her kind comments and try to amend the paper following his/her suggestions. Methods and results were better explained and the whole manuscript was revised by a Cambridge ESOL examination instructor to improve clarity, detect and correct potential grammar mistakes.

 Introduction

It’s not clear what this is saying. Edit for clarity “This travelers’ engagement with a country or region's culture enhances the potential of smaller cultural and heritage attractions to come out into business marketing all over the world [5], but also force living local communities to experiment substantial cultural and social changes, both positive and negative.”

Response: We rewrote the fragment to improve its clarity.

Methods

 “The total number of respondents was 131 (55 males, 76 females), that is the total number of 205 camel-riding consumers during the research period (1st July to 30th September 2019). Customer qualitative profile (age, nationality, study level and companions) is represented in Figure 1.” Please move this figure and it’s written description to the results section.

Response: Fragment and figure 1 were moved to results section.

“For this reason, we used a mixed-scale model to measure customer satisfaction and attitudes towards camel rides.” The use of the word model here is a bit confusing as I was expecting an econometric model. I think a more appropriate term would be approach.

Response: Changed to approach.

“Shapiro-Francia W' test (for 5≤n≤1000 samples) was performed to study data distribution using the Shapiro-Francia normality routine of the Stata Version 15.0 software.” I believe this is missing a citation.

Response: Citation was added.

“As provided parametric assumptions were grossly violated (Supplementary Table S1) and the strong subdivision occurring when testing for the effect of multiple factors may condition results, a Bayesian approach was followed to avoid power and precision loss in a relatively small data sample context.” This also needs a citation.

Response: Citations were added.

“Varimax with Kaiser Normalization Rotated Component was performed to reduce the number of factors to be included in our model.” Missing a citation

Response: Citation was added.

It is unclear to me how the Varimax with Kaiser Normalization Rotated Component is working with 2 different dependent variables customer general satisfaction and return intention probability. I think better interpretation of your findings would be helpful.

Response: We clarified in text.

In general- it would be helpful to the reader to introduce the main model first, and then explain the corrections done. I was unclear what the end-goal was for this method until I reached the section on OLS.

“Using significantly loaded factors for each of the six dimensions identified by PCA, we built simple linear regression models to predict for customer general satisfaction and return intention probability in regards to camel tourist walks using the Linear Regression package from the Bayesian statistics task of SPSS Statistics for Windows, Version 25.0, IBM Corp.” Please write out the model

Response: Methods section was rewritten to clarify the steps given and the purposes of each of these steps.

“The posterior predictive p values provide (PPP values) were computed as a goodness of fit measure for the model being tested.” It is unclear at this point what the model is?

Response: Information was provided to clarify models used in the text.

Results

“A summary of the descriptive statistics for the variables and potential conditioning factors of customer general satisfaction and return intention probability in regards to camel tourist walks, is included in Supplementary Table S2.” Space permitting- I believe this is an important contribution and should be included in the main paper.

Response: Descriptive statistics were included in the body text.

Table 1: The decision making factors- I am unsure of what it says as it is cut off- should be introduced previously in the methods. Also what do the grey boxes indicate? How are the numbers in the PC columns interpreted?

Response: We clarified and adjusted tables and clarified in the general section model. Interpretation for factor components loadings was also provided as a footnote.

Table 1- “Considering the variables clustered together on each principal component dimension” can you give an example of this. For example the clustered variables X, Y, Z led us to determine dimension 1 etc.

Response: We clarified this in text.

“Table 2 also suggests model comprising significantly loading factors in dimension 3 and model comprising significantly loading factors in dimension 6 were more likely to explain and predict for customer general satisfaction and return intention probability, respectively, than others comprising just the intercept.” It is difficult to interpret what is going on here without having the models written out. I believed you were doing OLS, but we never see coefficients that will help you make the connection between what is important to the visitors and the dependent variable with any kind of magnitude scale.

Response: As it is clarified in the text factors comprising dimensions 3 and 6 are the only ones presenting a significant linear relationship with the dependent variables of customer general satisfaction and return intention probability in regards to camel tourist walks. Linear regression Coefficients and the descriptive statistics for them (provided we followed a Bayesian approach) for these factors are provided in Supplementary Tables S4 and S5

“The posterior predictive p values for both models were around 0.5. This value indicates 396 moderately plausible good-fitting models.” It’s unclear what this is saying, and it needs a citation.

Response: Model validity section was explained and citations were provided for the parameters given.

Discussion

Table 3 belongs in the results section

Response: Table 3 was relocated.

“In fact, staff performance and trust-based animal-human relationships are potential features affecting customer satisfaction, according to our results (PC1, Table 1). Likewise, a mistrustful camel may be the result of a poor training workout, hence an indirect, negative judgment of previous staff performance and fair estimate of user satisfaction itself [53].” Potential features? Did the results show it had an impact or not?

Response: We modified the information in text to confirm the significance of the factors.

“Since user complacency was high in this regard (62 % of respondents were satisfied and highly satisfied)” this would be useful information to present in the results.

“Instead, the indirect estimation of respondents’ perception about pricing shows this factor is not as relevant as lengthiness sensory stimulation during the journey when explaining customer satisfaction.” I don’t see this in the results.

Response: We clarified and provided references. Relationship between price and quality-quantity can be inferred from global customer contentment on just mentioned characteristics (short, adequate or long journey, and whether the user suggests some improvement or not), basing on the role of price in quality and value judgments for consumer's satisfaction and return intention probability [1]. For instance, Homburg, et al. [2] reported that as satisfaction increases, the negative impact of the magnitude of a price increase is weakened.

Minor/editorial:

Pg 1 line 29- edit for clarity and length “Camels are iconic and exotic regional pull elements within adventure travel companies 29 that promote eco-tourism activities with strong potential to generate revenues for local livelihoods and promote educational values towards nature and its conservation.”

Response: Suggestion was followed and sentence was rewritten.

Pg 1 line 25- delete may “satisfaction and loyalty in this tourism segment and that may made it socially differentiate”

Response: Deleted.

“Bartlett's test of sphericity tests was performed to determine whether variables are unrelated and therefore unsuitable for structure detection.” Change to: Bartlett's test of sphericity was performed to determine whether variables are unrelated and therefore unsuitable for structure detection.

 Response: Changed.

“Initial and extraction communalities were assessed to determine which variables should be maintained of discarded from PCA.” I believe of should be changed to or

Response: We agree.

Reviewer 3 Report

This paper is very interesting and offers a novel approach to the literature about the relationships between animals, tourism and regional development. However, there are some issues the authors should revise prior to its potential publication:

- in the introduction the authors should provide further references to justify the reasons included in paragraphs between line 80 and 98. Also, the role of camels in tourism should be better described.

- theoretical framework needs to be reworked and contextualised. The authors can reorder the topics developed in order to show the process animal-territory-people-ethics. Literature about animal-based leisure and nature-based leisure should be better integrated into the discourse. Also, the relations between animal ethics and tourism should be further discussed.

- the authors should move the rationale of the study case to the beginning of the methodology, and provide a detailed description of the sample procedures and time distribution which will increase the credibility and validity of the study. Data about camel tourism demand would enrich the study.

-  also, how did previous literature inform the design of the questionnaire? In this sense, the review of consumer behaviour researches in domestic animal tourism should be used to complement the theoretical framework.

- the authors contextualise their discussion within the processes of consumption and decision-making, customer general satisfaction and return intention probability, which are not properly defined earlier in the paper. This should be revised.

- the improvement of the theoretical framework will contribute to expand the discussion, where the authors may consider to include a figure that summarizes the results and the implications, and illustrates the conceptualisation of a sustainable animal-based tourism.

Author Response

Reviewer 3

Comments and Suggestions for Authors

This paper is very interesting and offers a novel approach to the literature about the relationships between animals, tourism and regional development. However, there are some issues the authors should revise prior to its potential publication:

Resposne: We thank the reviewer for his/her kind comments.

- in the introduction the authors should provide further references to justify the reasons included in paragraphs between line 80 and 98. Also, the role of camels in tourism should be better described.

Response: Reviewer suggestion was followed.

- theoretical framework needs to be reworked and contextualised. The authors can reorder the topics developed in order to show the process animal-territory-people-ethics. Literature about animal-based leisure and nature-based leisure should be better integrated into the discourse. Also, the relations between animal ethics and tourism should be further discussed.

Resposne: Reviewer suggestion was followed and topics were reordered. Ethics and Tourism relationship was discussed.

- the authors should move the rationale of the study case to the beginning of the methodology, and provide a detailed description of the sample procedures and time distribution which will increase the credibility and validity of the study. Data about camel tourism demand would enrich the study.

Response: Information requested was provided.

-  also, how did previous literature inform the design of the questionnaire? In this sense, the review of consumer behaviour researches in domestic animal tourism should be used to complement the theoretical framework.

- the authors contextualise their discussion within the processes of consumption and decision-making, customer general satisfaction and return intention probability, which are not properly defined earlier in the paper. This should be revised.

Resposne: Concepts of customer general satisfaction and return intention probability were revised and redefined.

- the improvement of the theoretical framework will contribute to expand the discussion, where the authors may consider to include a figure that summarizes the results and the implications, and illustrates the conceptualisation of a sustainable animal-based tourism.

Response: Discussion was expanded following the suggestions to make on the theoretical framework of the manuscript and a figure to illustrate the conceptualization of camel rides as an example of sustainable animal-based tourism is provided as well.

Round 2

Reviewer 2 Report

The authors have adequately addressed all of my concerns.

Reviewer 3 Report

The authors have revised the paper accurately. However, animal sample changed from “14 dromedaries (9 males and 5 females; aged between 2 and 35)” to “8 dromedaries (6 males and 2 females; aged between 4 and 32)” and the authors should explain why.

Author Response

The authors have revised the paper accurately. However, animal sample changed from “14 dromedaries (9 males and 5 females; aged between 2 and 35)” to “8 dromedaries (6 males and 2 females; aged between 4 and 32)” and the authors should explain why.

  • Response:

The information in the sample provided was the total number of animals owned by the touristic Enterprise.

However, Reviewer 1 suggested Sample age range should be checked.

While revising this comment, we noticed out of the 14 dromedaries in the farm, only 8 had participated in the touristic rides considered in this study, hence we had to update the information.

We apologize for the misunderstanding.